# Exploring the Isomeric Precursors of Olive Oil Major Secoiridoids: An Insight into Olive Leaves and Drupes by Liquid-Chromatography and Fourier-Transform Tandem Mass Spectrometry

**DOI:** 10.3390/foods10092050

**Published:** 2021-08-31

**Authors:** Ramona Abbattista, Ilario Losito, Cosima Damiana Calvano, Tommaso R. I. Cataldi

**Affiliations:** 1Dipartimento di Chimica, Università degli Studi di Bari “Aldo Moro”, Via E. Orabona 4, 70125 Bari, Italy; ramona.abbattista@gmail.com (R.A.); tommaso.cataldi@uniba.it (T.R.I.C.); 2Centro Interdipartimentale SMART, Università degli Studi di Bari “Aldo Moro”, Via E. Orabona 4, 70125 Bari, Italy; cosimadamiana.calvano@uniba.it; 3Dipartimento di Farmacia e Scienze del Farmaco, Università degli Studi di Bari “Aldo Moro”, Via E. Orabona 4, 70125 Bari, Italy

**Keywords:** extra-virgin olive oil, olive leaves and drupes, secoiridoids, oleuropein, oleuroside, ligstroside, elenolic acid glucoside, secoxyloganin, liquid chromatography-Fourier transform tandem mass spectrometry

## Abstract

Secoiridoids play a key role in determining health benefits related to a regular consumption of extra-virgin olive oil (EVOO), in which they are generated from precursors of the same class naturally occurring in drupes and leaves of the olive (*Olea europaea* L.) plant. Here, reversed-phase liquid chromatography coupled to electrospray ionization and Fourier-transform single/tandem mass spectrometry (RPLC-ESI-FTMS and MS/MS) was employed for a structural elucidation of those precursors. The presence of three isoforms in both matrices was assessed for oleuropein ([M-H]^−^ ion with *m*/*z* 539.1770) and was emphasized, for the first time, also for ligstroside (*m*/*z* 523.1821) and for the demethylated counterparts of the two compounds (*m*/*z* 525.1614 and 509.1665, respectively). However, only the prevailing isoform included an exocyclic double bond between carbon atoms C^8^ and C^9^, typical of oleuropein and ligstroside; the remaining, less abundant, isoforms included a C=C bond between C^8^ and C^10^. The same structural difference was also observed between secoiridoids named elenolic acid glucoside and secoxyloganin (*m*/*z* 403.1246). This study strengthens the hypothesis that secoiridoids including a C^8^=C^10^ bond, recently recognized as relevant species in EVOO extracts, arise mainly from specific enzymatic/chemical transformations occurring on major oleuropein/ligstroside-like precursors during EVOO production, rather than from precursors having that structural feature.

## 1. Introduction

Extra-Virgin Olive Oil (EVOO) has been an essential component of Mediterranean diet for centuries. The progressive elucidation of its beneficial effects for human health has prompted its production and consumption worldwide in recent years [1]. Phenolic compounds have been reported to be significantly implicated in EVOO health benefits, with species belonging to the class of secoiridoids playing a major role among them [2,3,4,5,6,7,8,9]. Secoiridoids are key secondary metabolites of *Oleaceae* plants, including *Olea europaea* L., whose drupes are employed in olive oil production. Major compounds of this class in olive drupes and leaves, generally known as oleuropein and ligstroside (see Figure 1), arise from the formation of an ester linkage between phenolic alcohols known as 3-hydroxy-tyrosol (3,4-dihydroxy-phenylethylethanol, 3,4-DHPEA) and tyrosol (3-hydroxy- phenylethylethanol, 3-HPEA) and elenolic acid glucoside (also known as oleoside 11-methyl ester) [10,11]. Low concentrations of demethylated forms of oleuropein and ligstroside (see Figure 1) can also be found in olive drupes and leaves, although their abundance has been found to increase upon fruit maturation [12].

During the first steps of EVOO production, secoiridoids are involved in a complex network of enzymatic and chemical processes, occurring when olive drupes are crushed, and the resulting paste is subjected to the process known as *malaxation*. In particular, endogenous β-glucosidases [13,14,15] and methyl-esterases [16] play a fundamental role in transforming them into the corresponding aglycones and then into two further important secoiridoids of EVOO, i.e., oleac(e)in and oleocanthal.

Recently, a systematic structural investigation based on mass spectrometry was performed in our laboratory on major secoiridoids occurring in EVOO and in its processing wastes [17,18,19]. The coupling between Reversed-Phase Liquid Chromatography and Fourier-transform Mass Spectrometry with Electrospray Ionization (RPLC-ESI-FTMS) [17,18] showed that, once generated enzymatically, oleuropein and ligstroside aglycones are further transformed. In particular, an opening of the unstable cyclic hemiacetal initially included in the aglycones structure occurs, with a subsequent cyclization based on intramolecular 1,4-Michael addition. This can be followed by a new ring opening, triggered by acid conditions and able to generate di-aldehydic open-structure compounds [17,18]. As a result, more than 10 different isoforms, including also positional and geometric isomers, and even stable enolic tautomers, recognized by RPLC-ESI-FTMS after H/D exchange, were characterized for each of the original aglycones of oleuropein and ligstroside [17,18]. Interestingly, some of the most relevant isoforms were compatible with the displacement of the exocyclic C=C bond from carbon atoms conventionally numbered as 8 and 9, as in oleuropein and ligstroside, to those numbered as 8 and 10 (see Figure 1). This feature was reported in 1988 for a glucosidic secoiridoid named oleuroside (see Figure 1), whose structure was investigated by NMR spectroscopy after extraction from olive leaves [20] and was subsequently confirmed by Ryan et al. [10]. Notably, oleuroside should be originated by the formation of an ester bond between 3-hydroxytyrosol and a glucosidic secoiridoid named secoxyloganin (see Figure 1), one of the compounds involved in biosynthetic pathways of olive secoiridoids [11]. As apparent from the oleuroside structure reported in Figure 1, the displacement of the exocyclic C=C bond between C^10^ and C^8^ atoms makes C^9^ a new stereogenic center, potentially leading to two diastereoisomers upon coupling with the naturally fixed configuration of C^5^, the original chiral center of olive secoiridoids. Moreover, by analogy with the oleuropein/oleuroside couple, positional isomers of ligstroside (and eventually of demethyl-oleuropein/ligstroside) including the C=C bond between C^10^ and C^8^ might also be present in olive leaves and drupes, and thus play a role as precursors of some isoforms of secoiridoid aglycones in EVOO.

Notably, several papers have been published in the last two decades on the analysis of secoiridoids in olive leaves and/or drupes using mass spectrometry coupled to liquid chromatography [21,22,23,24,25,26,27,28,29,30,31,32]. In some cases, the occurrence of oleuroside in olive leaves or drupes extracts was explicitly indicated, based on the presence of a peak eluting later than that of oleuropein in the RPLC extracted ion current chromatogram referred to the [M-H]^−^ ion with nominal *m*/*z* ratio 539 [22,24,26,27,28]. However, no specific MS data were provided to confirm such assignment. Based on the similarity of the MS/MS fragmentation pathways, an additional peak, closer to that of oleuropein, was assigned as “oleuropein isomer” in three studies [25,26,27]. The presence of an isomeric form of oleuropein in extracts of drupes from different Spanish olive cultivars was recently reported also by Fernandez-Poyatos et al. [31]. Additionally, some differences in the profiles of product ions obtained for oleuropein (and its putative isomer) and oleuroside were evidenced by Quirantes-Piné et al. [27] after MS/MS analysis, yet an interpretation for this finding was not reported.

Starting from this background, the main goal of this work is a systematic structural investigation, based on RPLC-ESI-FTMS and MS/MS, on the isoforms related to oleuropein and ligstroside, to their demethylated forms and to elenolic acid glucoside and secoxyloganin in extracts of drupes and leaves collected from trees of a typical Italian olive cultivar, *Coratina*, whose oil is known to be rich in secoiridoids [33]. In particular, fragmentation pathways arising from MS/MS analyses will be carefully interpreted to evidence structural differences existing between isoforms.

## 2. Materials and Methods

### 2.1. Chemicals and Olive Drupes/Leaves Samples

LC-MS grade water, methanol and acetonitrile, HPLC-grade n-hexane, and standard oleuropein [2-(3,4-dihydroxyphenyl)ethyl-(2S-(2α,3E,4β))-3-ethylidene-2-(β-D-glucopyranosyloxy)-3,4-dihydro-5-(methoxycarbonyl)-2H-pyran-4-acetate] were obtained from Sigma-Aldrich (Milan, Italy).

Drupes and leaves samples subjected to the extraction of secoiridoids were collected from different trees of *Coratina* olive cultivar in an orchard located near the town of Trani (geographical coordinates: latitude 41.26138° N, longitude 16.39777° E), in the central part of the Apulia region (Southern Italy), where *Coratina* is the prevailing cultivar. Drupes and leaves were subjected to secoiridoid extraction within 24 h from collection.

### 2.2. Extraction of Secoiridoids from Olive Leaves and Drupes

Before being subjected to the extraction of secoiridoids, olive leaves were washed thoroughly with distilled water to remove dust and eventual contaminants and were then dried in a stove at 80 °C for two hours. Afterwards, the extraction was performed for 4 h, by immersing 1 g of leaves into 40 mL of LC-MS grade water heated at 80 °C, according to a variant of the procedure reported by Lee-Huang et al. [34]. In the present case, entire leaves were extracted, since preliminary experiments showed that cutting them into small pieces before extraction led to an anomalous increase of aglyconic secoiridoids, likely due to the action of endogenous β-glucosidase released upon leaf cutting. At the end of the extraction, residual leaves were carefully removed, and the supernatant was collected and subjected to centrifugation at 4500× *g* for 5 min to separate eventually suspended material. The extract was then stored at 4 °C until RPLC-ESI-FTMS analysis.

The extraction of secoiridoids from olive drupes was performed according to the procedure reported by McDonald et al. [35]. In particular, after freeze-drying, olives were pitted and then pounded in a mortar in order to obtain a small grain powder. A total of 10 g of the powder were put into a beaker with 50 mL of a CH_3_OH/H_2_O 1:1 mixture, covered with parafilm, and left for one hour at room temperature. The resulting extract was then washed with n-hexane, filtered through a nylon filter (0.45 μm porosity) and then stored at 4 °C until RPLC-ESI-FTMS analysis.

### 2.3. RPLC-ESI-FTMS Instrumentation and Operating Conditions

An Ultimate 3000 UHPLC system coupled to a Q-Exactive quadrupole-Orbitrap mass spectrometer (Thermo Scientific, Waltham, MA, USA) was employed for RPLC-ESI-FTMS analyses of extracts obtained from olive leaves or drupes. RPLC separations were performed using an Ascentis Xpress C18 column (150 × 2.1 mm ID, 2.7 µm particle size) preceded by an Ascentis Xpress C18 (5 × 2.1 mm ID) security guard cartridge (Supelco). A total of 5 μL of extracts were injected in the C18 column using the autosampler included in the Ultimate 3000 UHPLC system, equipped with a 6-way Rheodyne valve.

Separations were performed using the following elution gradient, based on water (solvent A) and acetonitrile (solvent B), already adopted in our laboratory for the separation of isomeric secoiridoids in olive oil extracts [17,18]: 0–5 min) 20% solvent B; 5–35 min) from 20 to 50% (*v*/*v*) solvent B; 35–40 min) from 50 to 100% of solvent B; 40–50 min) isocratic at 100% solvent B; 50–55 min) from 100 to 20% of solvent B; 55–70 min) column reconditioning at 20% solvent B. The flow rate was always set at 200 μL/min and the column temperature at 25 °C. Olive leaves/drupes secoiridoids were detected as [M-H]^−^ ions, generated upon deprotonation occurring during the ESI process. Specifically, ESI(-)-FTMS full scan acquisitions were performed in the *m*/*z* range 100–1500 after setting the main parameters of the heated ESI (HESI) interface and of the ion optics of the Q-Exactive spectrometer as follows: sheath gas flow rate, 60 (arbitrary units); auxiliary gas flow rate, 15 (arbitrary units); spray voltage, −4 kV; capillary temperature, 200 °C; S-lens RF level, 100 (arbitrary units). In the case of MS/MS analyses, the first isotopologue of each [M-H]^−^ ion of interest was isolated by the quadrupole analyzer of the Q-Exactive spectrometer (using a 1 *m*/*z* unit-wide isolation window) and then transferred into the Higher energy Collisional Dissociation (HCD) cell, after setting the Normalized Collisional Energy (NCE) as 20 (a.u.). Both MS and MS/MS analyses were performed by setting the spectrometer resolving power at its maximum value (120,000 at *m*/*z* 200). The spectrometer was calibrated on a daily basis using the Pierce^TM^ Negative Ion calibration solution (Thermo Scientific), containing sodium dodecyl sulfate (2.9 µg/mL), sodium taurocholate (5.4 µg/mL), and Ultramark 1621 (0.001%). The typical mass accuracy achieved was better than 5 ppm. The LC-MS instrumentation was controlled by the Xcalibur software (Thermo Scientific), used also for ion current extraction. The ChemDraw Pro 8.0.3 software (CambridgeSoft Co., Cambridge, MA, USA) was employed to draw chemical structures.

## 3. Results

### 3.1. RPLC-ESI(-)-FTMS Separation of Isomers of Oleuropein, Ligstroside, and Their Demethylated Forms in Extracts of Olive Leaves and Drupes

Recently, the characterization of secoiridoids in olive oil extracts by RPLC-ESI(-)-FTMS was successfully reported [17,18]. Here, the sample extracts of both olive drupes and leaves were investigated using the same analytical approach. Extracted ion current (XIC) chromatograms (extraction window width: 0.0040 *m*/*z* units) referred to exact *m*/*z* ratios of [M-H]^−^ ions of oleuropein, ligstroside, and of the corresponding demethylated forms (539.1770, 523.1821, 525.1614, and 509.1665, respectively), obtained after the RPLC-ESI(-)-FTMS analysis of extracts of leaves or drupes of *Coratina* cultivar, are reported in Figure 2A,B, respectively.

Minor chromatographic peaks were emphasized by magnifications reported in insets to panels b, c, d, e, and f of the figure. In agreement with previous findings [22,24,26,27,28,31], peaks related to isomeric forms were detected not only for oleuropein, but also for ligstroside and for demethylated oleuropein and ligstroside. Normalization Levels (NL) referred to the respective XIC traces suggested that demethylated forms were present at much lower concentrations (*vide infra*). Notably, some of the isomeric peaks were not detected in the XIC traces when drupe extracts were considered, likely due to the generally lower abundance of the four secoiridoids in drupes, also inferred from NL values reported in plots e-h of Figure 2B. From a chromatographic point of view, the reduction in hydrophobicity resulting from the demethylation of the -COOCH_3_ moiety of oleuropein and ligstroside led to a significant decrease in the retention times observed for their demethylated counterparts. On the other hand, oleuropein was eluted earlier than ligstroside and the same occurred for the respective demethylated forms, due to the higher polarity induced by the presence of two OH groups on the phenyl ring. To clarify the identity of each isomeric form related to peaks observed in XIC traces of Figure 2, HCD-MS/MS analysis was performed systematically on their [M-H]^−^ ions. A summary of chromatographic, mass spectrometric and structural information for all secoiridoids finally identified in extracts of olive leaves and drupes is provided in Table 1.

It is worth noting that estimates of concentrations for all compounds in the two matrices, expressed as mg/g of oleuropein, are also reported in the last column of the table, with the main purpose of comparing the incidence of specific isomeric forms in each type of sample. The estimates, expressed as average values referred to three replicates (with a maximum relative standard deviation of 14%), were obtained starting from the respective concentrations in the extracts, expressed as mg/L, and calculated using XIC peak areas retrieved for each isoform as responses and a calibration line obtained after the RPLC-ESI-FTMS analysis of standard oleuropein aqueous solutions in a 1–100 mg/L concentration range. Concentrations of secoiridoids in extracts were then transformed into concentrations in leaves or drupes (mg/g) by considering the amount of matrix and the volume of extracting solvent used in each case and assuming a quantitative extraction yield. It is worth noting that the use of oleuropein as a standard to estimate also the concentrations of ligstroside and demethyl-oleuropein/ligstroside assumes that these compounds have an ESI yield similar to that of oleuropein. This is reasonable, since the main negative ionization site of these secoiridoids, a OH group of the phenolic moiety, is the same as that of oleuropein. On the other hand, the ESI yields for secoxyloganin and elenolic acid glucoside are expected to be higher than that of oleuropein, since their negative ionization occurs on the COOH group, which is more easily deprotonated than the phenolic OH group. Concentrations reported for those secoiridoids in Table 1 might thus be slightly overestimated.

### 3.2. MS/MS Characterization of Isoforms of Oleuropein, Ligstroside, and of Their Demethylated Forms

#### 3.2.1. Oleuropein/Oleuroside

HCD-MS/MS spectra averaged under each peak detected in the XIC chromatograms obtained from olive leaves extracts and centered on the *m*/*z* ratio 539.1770, corresponding to the [M-H]^−^ ion of oleuropein, are reported in the corresponding plots of Figure 3A. It is worth noting that spectra were almost identical for the respective compounds in the case of leaves and drupes extracts. The close similarity of fragmentation patterns inferred for the three compounds despite the potential presence of structural differences recalled that observed for the different isomeric forms related to the aglycones of oleuropein and ligstroside [17,18]. The interpretation of MS/MS profiles was undertaken by considering the occurrence of oleuropein, as the prevailing species, and oleuroside, as already proposed in the literature [22,24,26,27,28,31]. As explained before, the displacement of the exocyclic C=C bond makes the C^9^ atom a stereogenic center in the case of oleuroside (see Figure 1). Due to the coupling between this center and the chiral center on C^5^, two diastereoisomers are expected for this secoiridoid, considering that, in principle, both configurations are possible on C^9^. Accordingly, two oleuroside-related chromatographic peaks should be detected in XIC traces obtained for the *m*/*z* value 539.1770, along with that of oleuropein (see Table 1). Although previous studies tentatively referred the second peak detected in XIC traces obtained for that *m*/*z* ratio to a not better clarified “oleuropein isomer” [26,27,28,31], a single isomeric form can be predicted for oleuropein. A careful interpretation of HCD-MS/MS spectra averaged under each chromatographic peak detected during this study helped us to clarify this issue.

As shown in Figure 1, the loss of dehydrated glucose, mediated by a 1,3-H transfer from a glucose carbon atom to the O atom linked to C^1^, led to the negatively charged cyclic hemiacetal of oleuropein aglycone (exact *m*/*z* ratio 377.1242), which was expected to be stabilized to open-structure isoforms corresponding to Open Forms I described in [17]. The resulting enolic-aldehydic and di-aldehydic open-structure isoforms, related each other through keto-enolic tautomerism, were reported in Figure 1. Not surprisingly, most product ions detected below the *m*/*z* value 377.1242 in the MS/MS spectrum of oleuropein (see the spectrum for peak #1 in Figure 3A) corresponded to those typical of oleuropein aglycone isoforms [17]. The structures of the most relevant ones, in terms of spectral abundance, among them (exact *m*/*z* ratios 345.0980, 307.0823, 275.0925, 223.0612, and 101.0244) were reported in the scheme. Notably, further product ions, compatible with exact *m*/*z* values 179.0561, 119.0350, and 89.0244, were detected in the MS/MS spectrum of oleuropein. As shown in the right side of Figure 1, they corresponded, respectively, to deprotonated glucose and to two fragments arising from it upon subsequent neutral losses of 1,2-dihydroxy-ethylene (reasonably stabilized to 2-hydroxy-ethanal upon keto-enolic tautomerism) and formaldehyde. Their generation clearly implied the presence of the negative charge on a glucose OH group, likely resulting from an intramolecular proton transfer occurring in the HCD cell before fragmentation (just one of the possible locations of the charge was shown in Figure 1), since the deprotonation of a glucose OH group during the ESI process appears much less likely. The product ion compatible with the exact *m*/*z* ratio 403.1246 could be explained by considering the negative ionization on glucose and the neutral loss of dehydrated 3-hydroxytyrosol, followed by a new intramolecular proton transfer, leading to a more stable ion, negatively charged on a COOH group. Notably, deprotonated glucose could be formed from the *m*/*z* 403.1246 ion, as depicted in Figure 1, or, eventually, even from the precursor ion, if deprotonated on the glucose moiety. In any case, the gas-phase generation of deprotonated glucose implies a complex fragmentation route in the case of oleuropein, since this compound has a C=C bond located between C^8^ and C^9^. Indeed, the only possibility for a breakage of the glycosidic bond with release of glucose is related to a H transfer from C^10^ to the glucose oxygen atom linked to C^1^, with subsequent displacement of the C^8^=C^9^ bond into the C^10^–C^8^ position.

As emphasized in the right side of Appendix A reported in the Appendix A, the entire process is expected to be much easier if oleuroside is considered, since the C=C bond is already in the C^10^–C^8^ position in this case, thus the detachment of deprotonated glucose simply requires a 1,3 H transfer from C^9^ to the glucose oxygen linked to C^1^. Moreover, as shown in the left side of Appendix A, the cyclic hemiacetal of oleuroside, formed upon neutral loss of dehydrated glucose from the precursor ion, can easily generate a product ion with exact *m*/*z* ratio 359.1136. This is expected to be a possible fragmentation route in the case of oleuroside, as an alternative to the opening of the cyclic hemiacetal discussed for oleuropein. The detection of distinct peak signals at *m*/*z* ratios compatible with the exact value 359.1136 in MS/MS spectra obtained for peaks #2 and #3, clearly more abundant than the one observed at that ratio in the MS/MS spectrum for oleuropein (see Figure 3A), can thus be considered a subtle but important hint to support the hypothesis that those peaks correspond to the two possible diastereoisomers of oleuroside.

#### 3.2.2. Ligstroside

As shown in panels A (plot b) and B (plot f) of Figure 2, three and two chromatographic peaks were observed, respectively, in the XIC traces obtained for the *m*/*z* ratio of deprotonated ligstroside after RPLC-ESI(-)-FTMS analysis of olive leaves/drupes extracts. This finding implied the detection of two further isomeric forms of this secoiridoid, not reported in previous studies [22,24,26,27,28,31]. The abundance of ligstroside isomeric forms was remarkably higher in the leaves extract, thus their MS/MS spectra were obtained from the RPLC-ESI-FTMS/MS dataset referred to this sample and are reported in Figure 3B. In this case, the level of similarity between fragmentation pathways was even higher than that observed for oleuropein/oleuroside. A peak referred to ligstroside aglycone (exact *m*/*z* 361.1293) was always observed, suggesting the detachment of dehydrated glucose, as discussed before for oleuropein. This process is described in Figure 1, where pathways referred to oleuropein and ligstroside were shown together. Note that exact *m*/*z* ratios referred to ligstroside product ions were reported in parentheses, like the H atom replacing the OH group on the *meta* position of the phenylic ring. As discussed in our previous work on ligstroside aglycones in EVOO [18], and as evidenced in Figure 1, product ions compatible with exact *m*/*z* ratios 291.0874, 259.0896, 223.0612, and 101.0244 were all related to deprotonated ligstroside aglycone, after it evolved from an unstable cyclic hemiacetal into a di-aldehydic or an enolic/aldehydic open-structure isoform, by analogy with oleuropein aglycone. Remarkably, product ions with exact *m*/*z* ratios 291.0874 and 259.0896 were generated, respectively, by neutral losses triggered by C^6^-to-C^9^ and C^6^-to-C^4^ 1,3-H transfers and the former ion was always the most abundant in the MS/MS spectrum, in accordance with MS/MS data reported by Quirantes-Piné et al. [27]. On the other hand, MS/MS spectra obtained for peaks #2 and #3 (see Figure 3B) shared an interesting feature, namely, the higher abundance, compared to the MS/MS spectrum obtained for peak #1, of product ions corresponding to deprotonated glucose (exact *m*/*z* ratio 179.0561) and its already described fragments with exact *m*/*z* ratios 119.0350 and 89.0244. Based on the same considerations made for the two isoforms of oleuroside, this spectral feature can be considered an important clue for the assignment of peaks #2 and #3 detected in the ligstroside XIC trace to the couple of diastereoisomers corresponding to positional isomers of ligstroside including a C^9^=C^10^ double bond (see Appendix A). As evidenced in Figure 2B, the diasteroisomer corresponding to peak #3 could not be detected in extracts of olive drupes, likely due to the generally lower content of secoiridoids in this sample. Peak #3 related to ligstroside or its positional isomers was less intense even in the case of leaves extract, compared to the corresponding peak in the oleuropein/oleuroside XIC trace (see Figure 2A).

#### 3.2.3. Demethyl-Oleuropein and Demethyl-Ligstroside

MS/MS spectra referred to the three peaks detected in the demethyl-oleuropein and demethyl-ligstroside XIC traces obtained for olive leaves extracts are reported in panels A and B of Figure 4, respectively. Fragmentation pathways proposed to explain major product ions detected in each case are reported in Figure 2, where, as in Figure 1, a double notation was used for *m*/*z* ratios of product ions not common to the two compounds, i.e., those including the phenolic moiety. It is worth noting that structures including the exocyclic C=C bond between C^8^ and C^9^, the position typical of oleuropein and ligstroside, were drawn in Figure 2. Among product ions shared by demethyl-oleuropein and demethyl-ligstroside, the one compatible with an exact *m*/*z* value 389.1089, generated by the neutral loss of dehydrated 3-hydroxy-tyrosol or dehydrated-tyrosol, according to the case, was prevailing in the case of demethyl-oleuropein isoforms. The process, mediated by a 1,3-H transfer, occurred when the negative charge was located on the COOH moiety, which is clearly the favored position in this case. As described in the left side of Figure 2, the product ion at *m*/*z* 389.1089 was subsequently involved in the neutral loss of CO_2_ (*m*/*z* 345.1191) from the COOH group generated by the phenolic moiety detachment. Product ions with exact *m*/*z* values 363.1085 and 347.1136, for demethyl-oleuropein and demethyl-ligstroside, respectively, were formed upon the usual loss of dehydrated glucose. They initially included the already described unstable cyclic hemiacetal, subsequently evolving into dialdehydic or enolic/aldehydic open structures depicted in Figure 2. The latter were able to explain the generation of several other product ions (detected especially in MS/MS spectra related to peaks #1, since they were the most abundant ones for both compounds) through neutral losses described in the scheme. Not surprisingly, fragmentation pathways leading to those product ions were generally similar to those previously observed for the aglycones of oleuropein and ligstroside [17,18].

As emphasized in Figure 2, further product ions could be explained by considering the deprotonation of a phenolic OH group, as a result of intramolecular proton transfer occurring on the precursor ion deprotonated on the COOH moiety. Indeed, after neutral loss of CO_2_, product ions with exact *m*/*z* ratios 481.1715 and 465.1766 were generated for demethyl-oleuropein and demethyl-ligstroside, respectively. The neutral loss of dehydrated glucose, leading to product ions with *m*/*z* 363.1085/347.1136, could be explained also when the negative charge was located on a phenolic OH group (see the right side of Figure 2). Notably, the direct neutral loss of the glucose molecule from the precursor ion (represented with a dashed arrow in Figure 2) could be considered to explain the generation of product ions compatible with exact *m*/*z* ratios 345.0980/329.1031, detected in MS/MS spectra of demethyl-oleuropein and demethyl-ligstroside (see Figure 4). The high resolution adopted for MS/MS analyses during this study enabled an easy distinction between ions with *m*/*z* ratios 345.0980 and 345.1191 detected for demethyl-oleuropein, although they cannot be seen as distinct peaks at the scale adopted for spectra in Figure 4. As explained before, the neutral loss of glucose implies a H transfer that is expected to be much easier in the case of secoiridoids including a C^10^=C^8^ double bond, instead of a C^8^=C^9^ one, thus it can be diagnostic for the actual structure of the precursor ion. Unfortunately, the generation of fragments with exact *m*/*z* 345.0980/329.1031 from negative ions of demethyl-oleuropein/demethyl-ligstroside may occur also through a different route, implying the loss of dehydrated glucose from precursor ions deprotonated on a phenolic OH group, followed by the neutral loss of water from one of the stable isoforms of the resulting aglycone (see Figure 2). Indeed, since a COOH group is linked to C^4^ in this case, it can undergo water loss and be transformed into a stable ketene (if it is not deprotonated).

None of the fragmentations observed upon HCD-MS/MS analysis of demethylated forms of oleuropein and ligstroside was thus able to provide a direct proof of the presence of positional isomers related to the location of the exocyclic C=C bond. Nonetheless, based on the strict similarity of chromatographic profiles (see Figure 2), the identities of the three isomeric forms of demethyl-oleuropein and demethyl-ligstroside detected in olive leaves extracts were inferred from those of their more abundant methylated counterparts. The most intense peak (#1) in XIC traces was thus considered to correspond to demethyl-oleuropein/demethyl-ligstroside including the C^8^=C^9^ double bond. Minor peaks (#2 and #3), one of which (#3) undetected for demethyl-ligstroside, were referred to the couples of diastereoisomers related to positional isomers including a C^10^=C^8^ double bond. Notably, peaks corresponding to these minor isomers were not detected in the XIC traces referred to the olive drupes extract (see Figure 2B). Their low abundance also in olive leaves extracts explains the relatively low S/N ratio affecting the corresponding MS/MS spectra, especially those referred to peaks #2, the least abundant isoforms (see Figure 4).

### 3.3. MS/MS Characterization of Elenolic Acid Glucoside and Secoxyloganin

Additional evidence for the presence of positional isomers related to the exocyclic C=C bond in secoiridoids of olive leaves and drupes was searched for by considering elenolic acid glucoside. As evidenced in Figure 1, this secoiridoid, also known as oleoside 11-methyl ester, corresponds to oleuropein missing the 3-hydroxytyrosol moiety, thus it includes a C^8^=C^9^ exocyclic bond. However, its positional isomer including a C^8^=C^10^ exocyclic bond has been also supposed to be present in olive leaves and drupes, and it is commonly known as secoxyloganin [10,11] (see Figure 1). Previous papers dealing with secoiridoids in olive leaves [26,27,28,31] reported the occurrence of one to four peaks in XIC chromatograms referred to the *m*/*z* ratio of deprotonated elenolic acid glucoside, thus suggesting that both types of compounds could be present. MS/MS data were also provided in some cases [26,27,28], but no structural interpretation was given, thus this aspect was considered in detail in the present work.

As shown in Figure 5, XIC traces obtained after the RPLC-ESI(-)-FTMS analysis of *Coratina* leaves (panel A) and drupes (panel B) extracts and referred to the exact *m*/*z* ratio (403.1246) expected for deprotonated elenolic acid glucoside or secoxyloganin, showed the presence of three major peaks, and three or two less abundant peaks, thus suggesting the occurrence of more isoforms than those reported so far. HCD-MS/MS spectra obtained for the three major peaks, shown in Figure 6, were interpreted to clarify the respective structures, considering that, in principle, one isomer was expected for elenolic acid glucoside and two diastereoisomers could be predicted for secoxyloganin (due to the already discussed two possible configurations at the stereogenic center represented by C^9^, see Figure 1). Several peaks detected in MS/MS spectra were in accordance with those found for elenolic acid glucoside in other papers, namely product ions reported with nominal *m*/*z* ratios 179, 223, and 371 in [26,28], and the ones reported with accurate *m*/*z* ratios 89.0235, 101.0229, 113.0239, 119.0347, and 165.0542 by Quirantes-Piné et al. [28]. Moreover, many of the detected product ions were already identified in this work when interpreting fragmentations of other secoiridoids found in olive leaves and drupes. A fragmentation scheme hypothesized to explain most signals detected in MS/MS spectra of Figure 6 is reported in Figure 3, considering the structure of elenolic acid glucoside for the precursor ion. As apparent, a neutral loss of methanol from the precursor, deprotonated on the COOH moiety, can be invoked to explain the generation of a product ion with exact *m*/*z* ratio 371.0984, which was detected for all isomeric forms, although it was more abundant for peak #1 (see Figure 6). The generation of elenolic acid, initially deprotonated on the COOH moiety (exact *m*/*z* 241.0718), could be easily explained considering the neutral loss of dehydrated glucose from the precursor, a process already discussed for oleuropein and ligstroside and for their demethylated counterparts. Even in this case, deprotonated elenolic acid was the origin of a complex network of fragmentation pathways, due to the possibility of breakage of the cyclic hemiacetal, with generation of an enolic/aldehydic open structure, able to displace negative charge also on the enolic OH group, through intramolecular proton transfer. As a result, the generation of product ions compatible with exact *m*/*z* ratios 223.0612, 165.0557, 139.0401, 113.0244, 101.0244, 71.0139, and 59.0139 could be explained. The set of product ions detected below *m*/*z* 200 was completed, at least for the isoform corresponding to peak #3, by product ions related to the negative ionization on a glucose OH group, i.e., deprotonated glucose (exact *m*/*z* 179.0561) and its fragments at *m*/*z* 119.0350 and 89.0244 (see the right side of Figure 3), with the latter being the most abundant. Based on its relative abundance in XIC traces of Figure 5, and on the excellent accordance between the related MS/MS spectrum and the fragmentation pattern shown in Figure 3, peak #3 was assigned to elenolic acid glucoside.

As apparent from Figure 6, only a part of product ions described so far was detected for peaks #1 and #2, whose MS/MS spectra were unique both for the detection of further, specific fragments, and for the negligible abundance of signals corresponding to deprotonated glucose and its fragments. Since two isomers were, in principle, expected for secoxyloganin, the interpretation of the two spectra was attempted by considering its structure for the precursor ions. The corresponding fragmentation patterns are reported in Appendix A. In this case, four specific product ions were fundamental to confirm the assignment of peaks #1 and #2 to secoxyloganin isomers. First, the fragment compatible with an exact *m*/*z* ratio 359.0984 could be interpreted considering the combination of a water loss from the glucose moiety (only one of the possibilities is shown in Appendix A) and a neutral loss of acetylene, explained through a 1,3 H transfer from C^10^ to C^9^ (see the top-left part of Appendix A). The latter process is clearly impossible in the case of elenolic acid glucoside, in which a double bond is located between C^8^ and C^9^. Interestingly, the generation of the product ion with *m*/*z* 359.0984 was more likely for peak #2 than for peak #1 (see the respective plots in Figure 6). Since the H transfer required for the acetylene loss is expected to be influenced by the configuration of substituents on C^9^, this result strengthens the hypothesis that peaks #1 and #2 in XIC traces of Figure 5 correspond to the two possible diastereoisomers of secoxyloganin, differing just for that configuration. Further, product ions detected in MS/MS spectra referred to the two peaks required the occurrence of a specific process to be explained. In particular, a key step was the opening of the dihydropyranic ring (see the right side of Appendix A), through a process that can be considered the reverse of the intramolecular 1,4-Michael addition described in our previous papers on oleuropein and ligstroside aglycones [17,18]. As a result, a double bond was generated between C^1^ and C^9^ and was fundamental for subsequent fragmentations, namely, those leading to other product ions unique for peaks #1 and #2, compatible with exact *m*/*z* ratios 245.1031, 227.0925, and 213.0769. As emphasized in Appendix A, these ions require the deprotonation of a glucose OH group as an intermediate step. The fact that deprotonated glucose and its fragments were not detected in the MS/MS spectra of peaks #1 and #2 indicates, indirectly, how favored was the opening of the dihydropyranic ring in this case. Moreover, as described on the left side of Appendix A, this process was also fundamental to explain the generation of product ions with exact *m*/*z* ratios 59.0139, 71.0139, 101.0244, and 113.0244, with the negative charge located on the COOH group or on an enolic OH group, according to the case.

Once peaks from #1 to #3 were identified, HCD-MS/MS spectra were tentatively considered also for minor peaks detected in the XIC trace referred to the exact *m*/*z* ratio 403.1246, eluted later than peaks referred to secoxyloganin diastereoisomers and to elenolic acid glucoside (retention times between 3.3 and 5 min, see Figure 5). Despite the lower S/N ratio of the corresponding spectra, a set of product ions similar to the one observed for elenolic acid glucoside was found for the three peaks, with none of the key product ions of secoxyloganin detected (data not shown). Based on this finding, the three/two additional compounds detected in the extracts of olive leaves/drupes might represent further isomeric forms of that glucoside, never considered so far. As a general hypothesis, the geometric isomerism related to the C^8^=C^9^ bond or the diastereoisomerism eventually resulting from an alternative configuration at C^1^ might be responsible for their generation. Unfortunately, MS/MS was unable to clarify these subtle structural features, thus a structure could not be assigned to minor peaks (the not numbered ones) detected in the XIC traces of Figure 5.

## 4. Discussion

The coupling between RPLC and high-resolution single and tandem mass spectrometry enabled a new, intriguing insight on major secoiridoids occurring in olive leaves and drupes, which represent the precursors of compounds of the same class acting as bioactive components in olive oil. For the first time, the occurrence of multiple isomers was evidenced not only for oleuropein, but also for ligstroside and for the demethylated counterparts of the two secoiridoids. A careful evaluation of the respective tandem MS spectra clarified that the major isoform for all of them included the exocyclic C^8^=C^9^ bond, a structural feature well known for oleuropein and ligstroside. Conversely, the less abundant isoforms always included a C^8^=C^10^ bond, a feature previously recognized in the literature only for the secoiridoid known as oleuroside, a positional isomer of oleuropein. MS/MS data suggested that the two diastereoisomers predictable for oleuroside were both present in olive drupes and leaves extracts. The same result was obtained, in the case of olive leaves, also for the two possible isoforms of ligstroside and of demethylated oleuropein/ligstroside including a C^8^=C^10^ bond, whereas one or both of those isoforms were not detected in drupes extracts. Additionally, the different location of the exocyclic C=C bond was assessed also for elenolic acid glucoside (C^8^=C^9^) and for the two isoforms of secoxyloganin (C^8^=C^10^), i.e., secoiridoids missing the phenolic moiety, detected both in olive leaves and drupes.

Notably, the presence of a C^8^=C^10^ bond in some isoforms of oleuropein and ligstroside aglycones was evidenced during previous analyses of Italian extra-virgin olive oil extracts based on the same RPLC-ESI-FTMS approach [17,18]. In many cases, those isoforms were found to be very relevant, or even prevailing over others, in terms of relative abundance, thus it is unlikely that they arise directly from oleuroside or ligstroside positional isomers, occurring in very low amounts in olive drupes. The results of the present work thus strengthen the hypothesis, made during those studies, that secoiridoid isoforms with a C^8^=C^10^ bond occurring in olive oil arise from a complex network of enzymatic and chemical processes involving the aglycones of oleuropein and ligstroside, originally including a C^9^=C^10^ bond. Specifically, the unstable cyclic hemiacetal characterizing these aglycones can be opened, with generation of an enolic/aldehydic species, that can subsequently undergo a cyclization through 1,4-intramolecular Michael addition [17,18]. The key step for the generation of secoiridoid isoforms with a C^8^=C^10^ bond is the subsequent re-opening of cyclic structures generated by 1,4- intramolecular Michael addition. Indeed, this process can either return open structures with the original C^9^=C^10^ bond or new open structures in which the C=C bond is located between C^8^ and C^10^ [17,18]. These reactions likely occur in the first stages of the olive oil production (especially during drupes crushing and malaxation) and might be modulated by the specific conditions adopted during those stages, thus explaining why the incidence of secoiridoid isoforms with a C^8^=C^10^ bond can change remarkably between different olive oils.

The results obtained during the present investigation provide useful indications also on the generation of two other important secoiridoids found in olive oil, i.e., oleac(e)in and oleocanthal, that correspond, respectively, to oleuropein and ligstroside aglycones both missing the COOCH_3_ moiety linked to C^4^. As inferred from Figure 1, these secoiridoids might arise, in principle, from demethyloleuropein and demethyligstroside upon loss of the COOH group linked to C^4^. However, the present study shows that the amount of the two demethylated secoiridoids in olive drupes is too low (see Figure 2) to account for the relevant concentrations of oleacin and oleocanthal that were found in olive oils [17,18]. Since enzymes with methyl-esterasic activity, i.e., able to catalyze the hydrolysis of the ester bond of the COOCH_3_ moiety linked to C^4^, have been recently evidenced in *Olea europaea* L. [16], it is reasonable to hypothesize that oleacin and oleocanthal may arise mainly from the aglycones of oleuropein and ligstroside upon enzymatically catalyzed breakage of the COOCH_3_ ester bond, followed by chemical decarboxylation of the resulting COOH group. Consequently, isoforms with a C^8^=C^10^ bond should be present in olive oil also for oleacin and oleocanthal. This hypothesis has been recently confirmed in our laboratory using the same analytical approach described in the present paper, and, notably, open-structure isoforms with a C^8^=C^10^ bond have been found to prevail in the chromatographic profiles obtained for the two secoiridoids [36].

The presence in olive oil of isoforms of oleuropein/ligstroside aglycones and of oleacin and oleocanthal including a C^8^=C^10^ bond opens interesting perspectives in terms of the overall bioactivity of major olive oil secoiridoids. Antioxidant properties are usually recognized for these compounds and ascribed to the presence of the phenolic OH group(s). On the other hand, as recently shown in our paper on secoiridoid alterations occurring upon prolonged storage of olive oil, their C=O groups are involved in oxidation to COOH groups, especially when open-structure isoforms are considered [36]. Such isoforms could then contribute significantly to the beneficial antioxidant properties of olive oil. Moreover, their di-aldehydic nature could be very important also in terms of cross-linking with proteins *in vivo*. Finally, as previously evidenced using RPLC-ESI-FTMS integrated by H/D exchange [17,18], secoiridoid open-structure isoforms with a C^8^=C^10^ bond are able to generate stable di-enolic tautomers. Assessing the role eventually played by these specific isoforms in determining the bioactivity of secoiridoids in olive oils could be a very interesting future development of the research on these compounds.

## 5. Conclusions

The profile of major secoiridoids occurring in olive leaves and drupes could be investigated in great detail using RPLC coupled with high-resolution single and tandem mass spectrometry. In particular, oleuropein was confirmed to be the most relevant secoiridoid in both matrices, accompanied by two minor species having the same molecular weight, but recognized as two diastereoisomers of oleuroside, a positional isomer of oleuropein including the exocyclic C=C bond between carbon atoms C^8^ and C^10^, instead of C^9^ and C^10^. The same structural feature was evidenced, for the first time, also for ligstroside and for demethylated forms of oleuropein and ligstroside. Moreover, it was recognized for elenolic acid glucoside and the two diastereoisomers of secoxyloganin, i.e., secoiridoids missing the phenolic moiety and considered as biosynthetic precursors of oleuropein, ligstroside, and their demethylated derivatives. The low abundance of secoiridoids with a C^8^=C^10^ bond in olive drupes represents an indirect confirmation of the hypothesis that isoforms with this structural feature, previously found to prevail in the profiles of oleuropein/ligstroside aglycones, oleacin, and oleocanthal in olive oil, arise mainly from major isoforms of oleuropein and ligstroside through a complex network of enzymatic and chemical processes, finally leading to the displacement of the exocyclic C=C bond.

## Data Availability

The data presented in this study are available on request from the corresponding author.

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
