# Peer review of "Exploring the Isomeric Precursors of Olive Oil Major Secoiridoids: An Insight into Olive Leaves and Drupes by Liquid-Chromatography and Fourier-Transform Tandem Mass Spectrometry"

_foods, 2021, doi:10.3390/foods10092050_

Round 1

Reviewer 1 Report

Thank you for the opportunity to review this article. 

The authors evaluated the Exploring the isomeric precursors of olive oil major secoiridoids: an insight into olive leaves and drupes by liquid-chromatography and Fourier-transform tandem mass spectrometry.

Here are my remarks

Line 39. Olea Europaea...Please write europaea

Line 141-145...Please provide the coordinates from the place of sampling

Line 162. CH3OH/H2O...Please write with the right way

Line 172. 5 L...is it possible? Please clarify

Please enhance the discussion section

Please add the conclusions section

Author Response

We thank the Reviewer for his/her useful suggestions and for pointing out the typing errors present in the first version of our manuscript.

All typing errors were corrected during the revision step. In particular, the Greek symbol for "micro" at line 172, originally placed before the L symbol, had been probably skipped during editorial formatting, thus we have added it again.

The coordinates for the place of sampling of olive leaves and drupes were added.

The discussion section was expanded to discuss how the results of the present paper can be related to those referred to two further important secoiridoids, i.e., oleacin and oleocanthal, that were detected during our previous studies on olive oil and found to include preferentially a C=C bond between carbon atoms 8 and 10. The paper describing how this structural feature was inferred has been added, as Reference #36, in the list of references.

The conclusion section has been added, as suggested by the Reviewer.

Reviewer 2 Report

The article entitled “Exploring the isomeric precursors of olive oil major secoiridoids: an insight into olive leaves and drupes by liquid-chromatography and Fourier-transform tandem mass spectrometry” provides valuable and new data on major secoiridoids occurring in olive leaves and drupes.

I have the following comments for this article

  1. The abstract is appropriate, the introduction adequate, and the material and methods have provided sufficient details.
  2. Line 364: Please correct the typological error (0.45 µm porosity) “µ” is missing
  3. At many places, the “m/z” is not presented in italics; please correct
  4. The results are presented well, especially the fragmentation pathways proposed in figure 1 (oleuropein and ligstroside), 2 (demethyloleuropein and demethylligstroside isoforms), and 3 (deprotonated elenolic acid glucoside) are very informative.
  5. Results can be further improved by providing the concentration (e.g., equivalent to oleuropein) of identified secoiridoids from olive leaves and drupes.
  6. Discussion: In the abstract, the authors mentioned that “Secoiridoids play a key role in determining health benefits related to a regular consumption of extra-virgin olive oil.” These health benefits can be included here in the discussion.

Author Response

We thank the Reviewer for his/her positive comments on our paper.

The typing error at line 364 has been corrected and all "m/z" have been reported in italics.

As suggested by the Reviewer, the concentrations of secoiridoids in olive leaves and drupes extracts were estimated in terms of mg of oleuropein equivalents per g of leaf/drupe, starting from a calibration of the ESI-MS response based on aqueous solutions of standard oleuropein. The resulting data were reported in Table 1 and a new paragraph was added to the paper to describe the quantification procedure.

The discussion section was expanded, introducing also new considerations on how the main health benefit of olive oil secoiridoids, i.e., their anti-oxidant properties, can be related to the molecular structures of their precursors in olive leaves and drupes.